**Data Availability Statement:** Revised Data Availability Statement: Data cannot be shared

# Evaluation of content validity and feasibility of the eVISualisation of physical activity and pain (eVIS) intervention for patients with chronic pain participating in interdisciplinary pain rehabilitation programs

**Elena Tseli**[1,2]*, **Veronica Sjöberg**[1], **Mathilda Björk**[3], **Björn O. Äng**[1,2,4,5], **Linda Vixner**[1,5]

**1** School of Health and Welfare, Dalarna University, Falun, Sweden, **2** Department of Neurobiology, Care Sciences and Society, Division of Physiotherapy, Karolinska Institutet, Huddinge, Sweden, **3** Pain and Rehabilitation Center, and Department of Health, Medicine and Caring Sciences, Linköping University, Linköping, Sweden, **4** Center for Clinical Research Dalarna—Uppsala University, Falun, Sweden, **5** The Administration of Regional Board, Department of Research and Higher Education, Region Dalarna, Falun, Sweden

* ezt@du.se

## Abstract

### Background

Chronic pain negatively influences most aspects of life, including aerobic capacity and physical function. The "eVISualisation of physical activity and pain" (eVIS) intervention was developed to facilitate individualized physical activity for treatment in interdisciplinary pain rehabilitation programs (IPRPs). The objective of this study was to evaluate the content validity and feasibility of the eVIS intervention prior to an effectiveness trial.

### Methods

In order to determine pre-clinical content validity, experts (n = 10) (patients, caregivers, researchers) participated in three assessment rounds using a Likert-scale survey where relevance, simplicity, and safety were rated, whereafter the intervention was revised. Item-content validity index (I-CVI), average, and overall CVI were used to quantify ratings. To determine content validity and feasibility in the clinical context, experts (n = 8) (patients and physiotherapists) assessed eVIS after a 2-3-week test trial, with the feasibility aspects acceptability, demand, implementation, limited efficacy-testing, and practicality in focus. Additional expert interviews (with physiotherapists, physicians) were conducted on two incomplete areas.

### Results

The intervention was iteratively revised and refined throughout the study. After three assessment and revision rounds, the I-CVI ratings for relevance, simplicity, and safety ranged between 0.88 and 1.00 (≥0.78) in most items, giving eVIS "excellent" content validity. In

publicly because of ethical restrictions involving human research participant information. Data contains potentially identifying information and sensitive patient information. Data are collected from a small group of participants and especially qualitative data, such as free-text comments in questionnaires and diary notes, risk compromise participant privacy. Data is only available from the Dalarna University Institutional Data Access for researchers who meet the criteria for access to confidential data, after approval from the principal investigator. Data requests may be sent to the Dalarna University Institutional Data Access, dataskydd@du.se. De-identified individual data (participant ratings) used in this publication will be made available on reasonable request to the principal investigator. However, such a request must lie within the limits of the ethical approval for this project. In addition, to uphold data safety and other legal aspects, relevant agreements must be established prior to data being available outside the research group.

**Funding:** Funding This work was supported by the Swedish Research Council for Health, Working Life and Welfare [L.V., grant number 2017-00491, https://forte.se/]; the Research Council [L.V., grant number 2018-02455, https://www.vr.se/]; the Swedish Association for Survivors of Polio, Accident, and Injury [L.V., grant number 2020-03, https://rtp.se/], and research funding from Dalarna University [L.V., No grant number, https://www.du.se/]. The funders had no role in study design, data collection and analysis, decision to publish, or preparation of the manuscript.

**Competing interests:** The authors have declared that no competing interests exist.

the IPRP context, the intervention emerged as valid and feasible. Additional interviews further contributed to its content validity and clinical feasibility.

## Conclusions

The proposed domains and features of the eVIS intervention are deemed valid in its content and feasible in the IPRP context. The consecutive step-by-step evaluation process enabled careful intervention development with revisions to be made in close collaboration with stakeholders. Findings implicate a robust base ahead of the forthcoming effectiveness trial.

## Introduction

Chronic pain, defined as pain that persists or recurs for more than three months [1], is a prevalent and complex condition that affects nearly one in five people globally [2]. Recent definitions acknowledge that many of the common musculoskeletal pain conditions are increasingly conceived as 'a disease in their own right', debuting either as chronic primary pain or chronic secondary pain when caused by an underlying disease [1]. Chronic pain has negative effects on numerous aspects of life and impacts on physical and emotional functioning as well as quality of life [2–5]. Typically, aerobic capacity and functional levels are gradually reduced, leading to physical activity and exercise becoming essential treatment components [6]. Interdisciplinary Pain Rehabilitation Programs (IPRPs), a type of Interdisciplinary Treatment [7], are recommended to target the physical, emotional, and social consequences of this complex disease [3]. However, at group level, effectiveness studies reveal suboptimal results on major outcomes such as physical and mental health, pain intensity, and sickness absence [8, 9]. Physical activity and exercise positively influence quality of life, activities of daily living, pain intensity, and overall physical function, and they also reduce the risk of social isolation. For these reasons, they are considered core features of IPRPs [6, 10, 11]. However, low physical activity levels and the proper dosage and pacing of physical activity are recognized as being difficult to reform in the treatment of chronic pain. Possible explanations for this include factors such as depression, high pain intensity, and fear of movement [12–14]. There is a need for additional guidance to optimize individuals' physical activity levels based on their prerequisites and goals.

To facilitate the individualization of physical activity levels, we developed a novel eHealth intervention named eVISualisation (eVIS), for physical activity and pain. eVIS combines an easily accessible, commercially sold, wrist-worn activity tracker with a specifically designed web-application named PAin and TRaining ONline-web application (PATRON), which is designed to target facilitating techniques for behaviour change such as outcome expectations, self-monitoring, and self-evaluation, all of which are theoretically framed by Social Cognitive Theory [15, 16]. It has been found that the effectiveness of health promoting interventions increases when one or more behaviour change techniques are included, such as self-monitoring, goal setting, feedback, and review of achieved goals [17, 18]. These intervention components are assumed to influence behaviour, through changes in knowledge and awareness, for example, or beliefs of capability, and motivation [18, 19]. eVIS is intended to be used as an additional clinical tool within the IPRP context. Objective monitoring of physical activity combined with self-reported data enables the unique adaptation of physical activity prescriptions based on individual barriers and resources. The effectiveness of eVIS is to be evaluated through a registry-based randomised controlled clinical trial (R-RCT), described in the study protocol by Sjöberg et al. [20].

Efforts have been made to develop health-promoting interventions in the treatment of chronic pain as well as in adjacent fields [18, 21–24]. However, as crucial information on the content validity and feasibility of interventions often remains unknown, effect or effectiveness trials are impacted negatively [25, 26]. The updated Framework for Developing and Evaluating Complex Interventions divides the development of and research investigating complex interventions into four-phases: development or identification of the intervention, feasibility, evaluation, and implementation. In all phases, researchers aim to identify key uncertainties in the intervention's core elements: its context, key components of the intervention, stakeholders, key uncertainties, intervention refinement, and economic considerations [27]. Validity, feasibility, and acceptability levels often remain as uncertainties after the developmental phase, favourably investigated in the feasibility and evaluation phase [27]. In the field of eHealth, an intervention's content validity may be defined as "the extent to which its intervention activities are relevant to the underlying construct (i.e., program theory) and likely to be effective in achieving a particular intervention purpose in the intended population" [28]. Aside from assuring content validity, knowledge of the intervention's potential to be implemented in its intended context is urgent as it potentially affects effectiveness trials. It is therefore recommended that an intervention's feasibility is evaluated [26, 29]. In the initial developmental process of eVIS, we evaluated the criterion validity of the objective measurement of physical activity (steps) by activity tracking with Fitbit Versa [30]. Simultaneously, the eVIS-intervention was developed in collaboration with stakeholders and a software team (Nordforce Technology AB). However, the intervention's content validity and feasibility remained unknown. We hypothesized that the newly developed eVIS would be a valid and feasible intervention to facilitate physical activity in IPRPs. Consequently, before proceeding to an effectiveness trial, eVIS's content validity and feasibility in its intended context must be evaluated.

## Objectives

The overall objective was to develop and evaluate the content validity and feasibility of the eVIS intervention prior to an effectiveness trial.

1. To evaluate the content validity of eVIS in a pre-clinical context

2. To evaluate the content validity and feasibility of eVIS in a clinical context

## Methods

### Study design and context

An observational design with expert assessments was used to evaluate the content validity and feasibility of eVIS as a supplement to IPRPs [29, 31–33]. The methodology drew on a systematic process, including iterative development and quantification of subject expert judgment as a ground for evaluation, a method previously used for validation in instrument development [31, 34] but also tested for intervention protocol by our research group [33]. A consensus panel, constituted by another panel of experts, would then review and evaluate the expert judgments to identify and agree upon necessary revisions to improve the intervention protocol.

The study was performed in two consecutive steps; step 1 targeted objective 1 and step 2 targeted objective 2. Step 1 was a pre-clinical evaluation of the content validity of eVIS where iterative expert assessments of the eVIS-intervention protocol with consensus-panel revisions after each loop were conducted. Experts were recruited from four Swedish health-care regions of varying size. Step 2 was an evaluation of the content validity and feasibility of eVIS in a clinical context, where caregivers and patients tested eVIS in real life at three IPRP clinics within specialized and primary care, representing larger and smaller communities in Sweden (Fig 1).

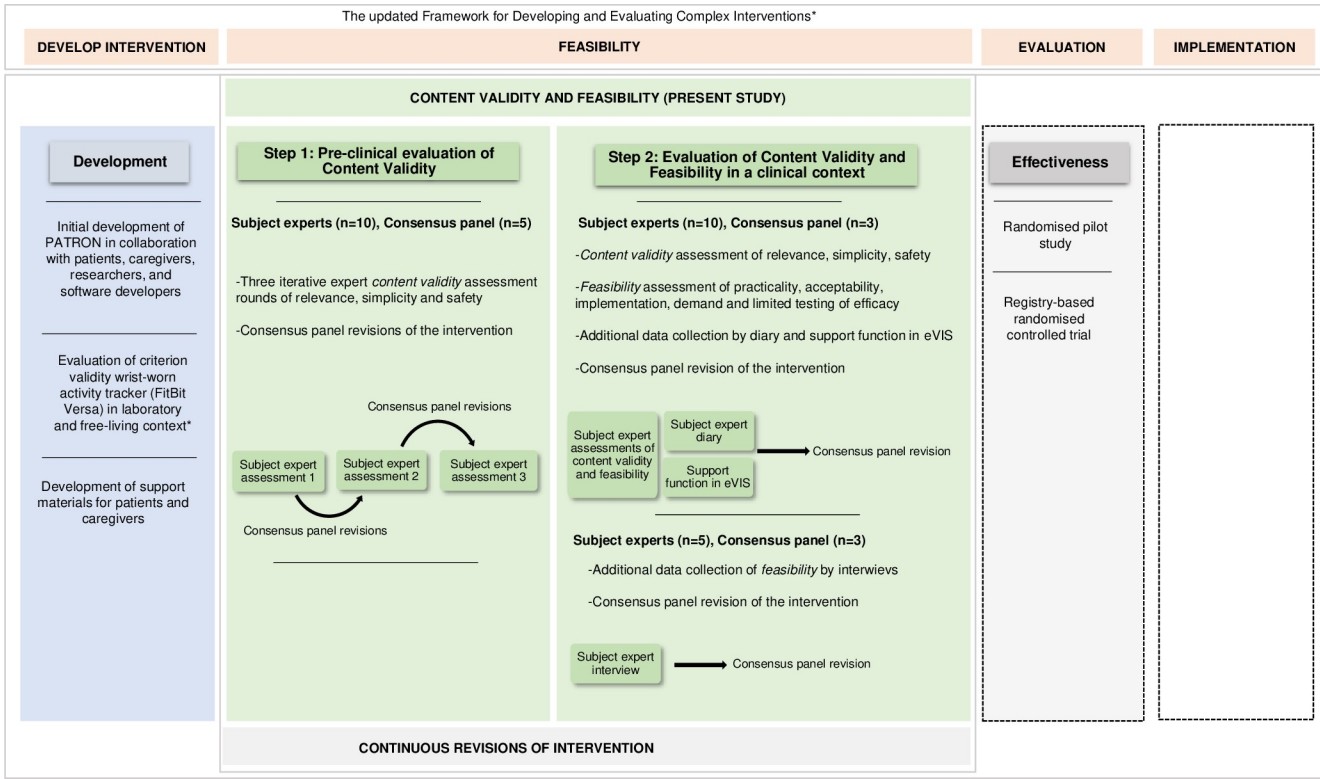

**Fig 1. Development, content validity, and feasibility evaluation stages of the eVIS intervention according to the updated Framework for Developing and Evaluating Complex Interventions.** * [35] Broken lines illustrate the subsequent phases with effectiveness evaluation and potential implementation.

The study was approved by the Swedish Ethics Review Board in June 2020 (Dnr. 2020/02033).

## Subject of evaluation—The eVIS intervention

eVIS is a new intervention. Details of the intervention and the forthcoming effectiveness trial are described in a study protocol by Sjöberg et al. [20] In short, the eVIS intervention consists of three elements: *Data collection*, *Visualization*, and *Communication*, see Fig 2 for a schematic illustration.

*Data collection*: The specifically designed web-application PATRON integrates real-time data collected from the activity tracker (steps/day) with daily patient-reported data of relevance to physical activity in chronic pain, all of which are recommended outcomes in clinical trials

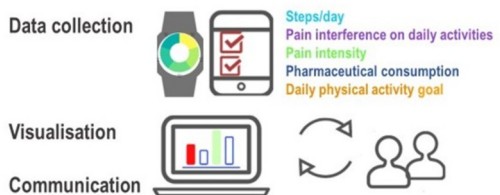

**Fig 2. Schematic overview of the eVIS intervention's elements.** Copied from study protocol by Sjöberg et al. [20]. Attribution CC BY-NC 4.0.

of chronic pain treatments [36, 37]: pain intensity, pain interference with daily activities, and pharmaceutical consumption. Numeric scales are used to assess pain intensity ("Mark the number corresponding to your average pain over the last day", 0 = no pain at all, 10 = worst imaginable pain) [38] and pain's interference with daily activities* ("Mark to what extent your daily activities are affected by your pain over the last day", 0 = not at all, 10 = to a very large extent). The latter is an adaptation of question two in the West Haven-Yale Multidimensional Pain Inventory (MPI) [39] (*replacing pain self-efficacy (PSEQ-2) [40] included in the initial version). In addition to numeric scales, a pharmaceutical report function is included for daily reporting (name, dose, quantity, and form). *Visualization*: Data is graphically visualized in PATRON (1/7/28 days) in relation to the patient's daily activity goal. When meeting with the clinician, the patient can log in and show the graphs. Based on data in PATRON, the patient and IPRP team get unique, detailed trends of information that provides possibilities for communication on barriers and facilitators of physical activity and to fine tune individualized treatment in a rehabilitation context. The caregiver and the patient together decide how to use and integrate the clinical tool as fits them.

## Subject experts and consensus panel

Subject expertise was the criterion measure used to determine the intervention's content validity and feasibility. Subject expertise constitutes an established knowledge base, where participants' professions or first-hand lived experiences represent the unique understanding of the subjects of interest [31]. The choice and range of the desired expertise depends on the objective, however including representatives of the target population is considered an important component, due to their familiarity with the construct through direct personal experience [31]. Hence, eligibility criteria for subject experts were that they had to be patients, caregivers, or researchers with knowledge and experience of chronic pain, interdisciplinary pain rehabilitation and IPRPs, research in physical activity, eHealth interventions, and patient-reported outcomes. Experts were strategically recruited through communications with patient organisations, colleagues, and the Swedish Quality Registry for Pain Rehabilitation network (SQRP), until the expertise needed was fulfilled.

**Step 1—Pre-clinical evaluation of content validity.** For step 1, **subject experts** (n = 10) were recruited to form a subject expert group to evaluate eVIS's content validity [31, 41]. A strategic recruitment of study participants was conducted to achieve a broad representation of the eVIS target population, combined with documented expertise that was either clinical or research-based within the fields relating to eHealth, physical activity, behavior change, outcome measures etc. The expertise area was verified by the initial questions in the questionnaire used for Assessment 1. Based on literature recommendations, the number of experts was set à priori to a minimum of 10, to avoid the risk of chance agreement [32, 41]. The group consisted of patient representatives (n = 4), experienced clinicians at specialist and primary care IPRP units (n = 4), and researchers (n = 2). Some were both clinicians and researchers. Patients were representative of the target population and had a chronic pain diagnosis, three of them for more than 5 years and one between 2–5 years, and experience of interdisciplinary pain rehabilitation. The clinicians and researchers specialized in IPRP; most of them had over five years of clinical experience from the targeted intervention. Researchers had research experience in the fields of pain and rehabilitation, physical activity, validation studies, implementation research and technical innovation, and eHealth. Five of the experts (both patients and clinicians) had previous experience of using activity trackers for either personal use or research.

**Step 2—Evaluation of content validity and feasibility in a clinical context.** In step 2, **subject experts** (n = 10) provided real-life experience data on their trial of eVIS. The subject experts were specialized physiotherapists (n = 3) from three different clinical settings across Sweden and a convenience sample of patients (n = 7) who were recruited within the IPRP setting (internal allocation was 1:1, 1:3, and 1:3). In the ethical permit there was an à priori decision to include one to three clinics, each including one to three patient participants. Recruitment of subject experts in step 2 was partly hindered because of reduced activity in tertiary care due to the ongoing Covid-19 pandemic. Patients were recruited based on the following criteria: accepted onto an IPRP due to non-malignant, musculoskeletal, or widespread pain for more than three months, aged 18–67, good ability to comprehend information and instructions given in Swedish, and daily access to a web browser using a computer, smart phone, or tablet. Patients in need of walking aids were not included. Two of the patients had previous experience of using activity trackers.

By the end of step 2, further expert interviews were needed regarding two areas emerging as incomplete in previous assessment processes (recruitment logistics, and pharmaceutical report function). Therefore, the physiotherapists were again invited as subject experts in a follow-up interview, of which two were able to participate. Additionally, subject experts (n = 3) with experience in pharmaceutical pain management of the specific population were recruited though our research group to validate specifics of the pharmaceutical report function. They were experienced physicians and researchers representing different regions in Sweden, and they participated in video-interviews.

All experts provided informed consent before taking part in the study and were told that they could withdraw at any time without giving a reason. Confidentiality was protected throughout the project. Data was stored at a project-specific server at Dalarna University, which prevented unauthorized access and was regularly backed up.

**Consensus panel.** A **consensus panel** handled the results from subject expert assessments and detailed the revisions. In step 1, the panel consisted of five health researchers (the author group) and one researcher specializing in visual perception. The competence in the consensus panel included: chronic pain and IPRP evaluation, physical activity and activity measurement, experience in epidemiological and experimental investigations, questionnaire and intervention development, eHealth and audiovisual communication, design, and production. In addition to the researchers, Nordforce Technology AB was present at all panel meetings to consult on technical and user-experience revision aspects. In step 2, a core consensus panel consisting of three of the researchers from author group remained throughout all steps of the study, but could, if deemed necessary, seek external specific expert advice, for consultation before specific revisions.

Hence, subject experts and consensus panel covered a multitude of expertise of relevance for the study's objective, e.g., experience of living with chronic pain, clinical pain rehabilitation, research in physical activity, research of patient-reported outcomes, image production, and software development.

## Outcomes

**Step 1—Pre-clinical evaluation of content validity.** To determine important aspects of content validity, a 4-point ordinal rating scale is commonly used [31, 32, 41]. It can be quantified to Content Validity Index (CVI), signifying the expert raters' proportion of agreement on an item's validity. The content validity of eVIS was evaluated based on three validity aspects: relevance, simplicity, and safety [31–34, 42, 43]. Relevance here refers to the extent that the proposed intervention and its included elements seem relevant as a means for facilitating

physical activity and improved physical health in the specific population. Similarly, simplicity refers to whether the intervention is easy to conduct, the clarity of instruction etc., and safety refers to whether any potential risks can be identified in using the intervention, for instance the risk of adverse events. Within the context of instrument development, the relevance and simplicity of items constitute common validity aspects for rating, but when used in the context of an intervention validation, safety is of equal significance, and has previously been used within our research group [33].

The relevance, simplicity, and safety was evaluated for the three elements of eVIS: Data collection, Visualization and Communication, and for the eVIS intervention in its entirety using a four-point Likert scale ranging from 1 to 4; 1: Not relevant/simple/safe, 2: Needs revision, 3: Relevant/Simple/Safe but needs minor alteration, 4: Very relevant/simple/safe. The Content validity index (CVI) was then used as quantitative indicator of content validity, operationalized as item-level CVI (I-CVI), scale-level CVI average (S-CVI/Ave), and scale-level CVI average/ domain (S-CVI/Ave Domain) [31, 32].

**Step 2—Evaluation of content validity and feasibility in a clinical context.** The content validity of eVIS in a clinical context, step 2, was evaluated based on the same three validity aspects as used in step 1: relevance, simplicity, and safety. The feasibility of eVIS, step 2, was evaluated based on five focus areas proposed by Bowen [29] to answer the overarching question "Can it be done?" These focus areas were: practicality (i.e., to what extent the intervention can be carried out with intended participants using existing means, resources without outside intervention), acceptability (i.e., to what extent the intervention is judged as suitable, satisfying or attractive to intervention deliverers and recipients), implementation (i.e., to what extent the intervention can be successfully delivered to intended participants in some defined but not fully controlled context), limited efficacy testing (i.e., does the intervention show promise of being successful with the intended population), and demand (i.e., to what extent the intervention is likely to be used?) [29]

Feasibility of the eVIS intervention in its entirety and its domains was assessed using a four-point Likert scale ranging from 1 to 4, (1: Not at all, 2: To some extent, 3: To a rather large extent, 4: To a large extent). The last item in the patient and physiotherapist questionnaires (Item P-F17 and PT-F23 respectively) was rated 1–4 to reflect the extent of agreement with a statement; 1: Strongly disagree, 2: Disagree, 3: Agree, 4: Strongly agree.

## Data collection and assessment processes

**Step 1—Pre-clinical evaluation of content validity.** A web-based questionnaire was used for the expert group to independently assess the content validity of the eVIS intervention in three assessment rounds, every second week over a period of six consecutive weeks. Three rounds of assessment were planned a priori to enable repeated input and enough time and resources for an iterative development process, with revisions made between rounds and modifications again evaluated. Prior to the first assessment round, the rationale for the validation study and the validation process were presented to the participating experts through an online video-meeting along with a brief introduction of the eVIS and designated webpage, for the experts to reference during assessment.

In the questionnaire the eVIS intervention in its entirety plus its three elements (hereafter together referred to as domains) were presented in text, photos, and figures followed by 28 questions (in analyses referred to as content validity items), which were rated regarding relevance, simplicity, and safety, amounting to 78 ratings (for seven items the safety dimension was irrelevant and therefore not included). In addition to every quantitative rating there was a commentary field for additional free-text comments. The questionnaire concluded with three

open-ended questions asking whether any part of eVIS should be excluded, or any additional part should be included, and whether there were any safety aspects that the experts wanted to highlight.

Between assessment rounds, ratings and free-text comments were examined by a consensus panel and revisions were made before the following assessment. The consensus panel evaluated the results and agreed on a priority list, sorted in a "*need to change*" or "*nice to change*"- order. When possible, within economical restrictions and timeframes, as many of the "*nice to change*" suggestions were also taken into account, as it was evident these would be of value. In the second round, the questionnaire consisted of only 20 items as eight items had been omitted, so only those that had had revisions made were rated. In the third round the full questionnaire was again rated.

**Step 2—Evaluation of content validity and feasibility in a clinical context.** To enable real-life clinical evaluation of the feasibility of eVIS within IPRPs, the intervention was implemented in its intended setting and tested for 2–3 weeks. The implementation was supported by a designated webpage with instructions.

Multiple data sources were used for the assessment: First, an analogue diary was used to record physiotherapists' and patients' momentary notes on experiences during the start up and trial of eVIS. Any telephone calls and e-mails from participants directed to the support function of eVIS were also used as a source of information at this step. Secondly, after the testing period had ended, two different web-based questionnaires (one for patients and one for physiotherapists) were used to assess the content validity and clinical feasibility of eVIS, as perceived from the participants' practical experience of testing it. Five items from the content validity questionnaire, previously used in step 1 –evaluation–were included (C11, C14, C19, C24, and C25 –see items in Table 2); these were three overall items for rating the domains Data collection, Visualisation and Communication, and eVIS objective in its entirety, and one item specifically targeted at the pharmaceutical report function in PATRON (C11). These items were identical in both questionnaires and were evaluated by I-CVI. The feasibility questions (feasibility items, F) were diversified according to respondent category (patient or physiotherapist) and covered the domains and overall implementation of eVIS from patient and caregiver perspectives, resulting in 17 and 23 items respectively (due to the latter being complemented with specific items on recruitment and the start up process). Feasibility items addressed the perceived feasibility of eVIS in relation to our chosen focus areas; acceptability, demand, implementation, limited efficacy, and practicality. All items had ratings and commentary fields.

Following finalization, exports of participant data records in PATRON were used to validate the export procedure (to pilot the functionality and accuracy of the eVIS data registration process and subsequent data export for analysis, i.e., testing of the entire data collection step).

Finally, video interviews were used for additional assessments on two specific areas as a need for immersed evaluation was identified during the analysis process. Details of the recruitment and initiation procedures (recruitment logistics) were qualitatively reviewed by physiotherapists, who participated in a group video interview. In addition, three individual video interviews with physicians with expertise in clinical pain management pharmacology were carried out, addressing the comprehensiveness of the pharmaceuticals list provided in PATRON as well as the rating of content validity of the pharmaceutical report function (item C11 from previous questionnaires) by means of physicians' medical expertise.

## Calculations/Analyses

**Step 1—Pre-clinical evaluation of content validity.** An Item-level CVI (I-CVI) was calculated for every validity aspect of every item, with values ranging from 0 to 1. In accordance with recommendations by Polit et al., an I-CVI $\geq 0.78$ was selected as cut-off for an excellent

I-CVI [32]. As a measure of inter-rater agreement on the overall relevance, simplicity, and safety of eVIS, a scale-level CVI average (S-CVI/Ave) was calculated. S-CVI/Ave implies the average I-CVI across all items in the questionnaire, (sum of all I-CVIs divided by number of I-CVIs) and respectively a S-CVI/Ave Domain, signifying the average I-CVI across items in a domain (eVIS data collection, visualization, communication, eVIS as a whole). A S-CVI/Ave of ≥0.9 was considered to be "excellent" content validity. Additionally, the Scale-level CVI universal agreement (S-CVI/UA) was used to evaluate agreement between experts (sum of all I-CVIs equal to 1 divided by number of I-CVIs) and shows the proportion of items that attained a rating of ≥3 from all the experts, with ≥0.8 considered as excellent [31, 32]. Free-text comments adjacent to the item ratings were interpreted at manifest level, close to the content, and supplemented the interpretation of the ratings.

**Step 2—Evaluation of content validity and feasibility in a clinical context.** The content validity in a clinical context was analysed based on I-CVIs from five items, calculated in the same way as in step 1, and adjacent narrative comments.

Quantitative data from the feasibility items in the patient and physiotherapist questionnaires were analysed descriptively and reported as the frequency and range of ratings for every item. The feasibility of an item was considered satisfactory if ratings were ≥3 (3: To a rather large extent/Agree or 4: To a large extent/ Strongly agree). To structure the investigated aspects of feasibility, items were categorized based on Bowen's suggested focus areas: acceptability, demand, implementation, limited efficacy testing, and practicality [29]. The categorization of feasibility items was performed independently and blinded by two of the researchers, and any disagreements on categorization were discussed and resolved through consensus. Free-text comments adjacent to ratings in questionnaires complemented the interpretation. Qualitative data collected from the diaries, e-mails, and support function were examined with a focus on identifying areas in need of attention with regards to feasibility and avoidance of adverse events.

**Handling of missing data.** No specific analyses of missing data were performed in this study. Reasons for missing data were noted if known and presented. In step 1, the I-CVI calculations in the final round were based on eight respondents (of ten) due to two of the participants not being able to attend. Also, in items C22, C24, and C25, there were occasional missing data as one of the respondents declared in the text field that it was not possible to rate these items as there was no option to omit a rating, leading to an arbitrary number being chosen. I-CVI calculations were in this case recalculated with a missing value instead and with initial number respondents in the denominator. In the evaluation of feasibility, in step 2, there was partial missing data on questionnaires and diaries. Two of the participants did not respond to the questionnaire due to sickness or an unknown reason. Two of the diaries were not submitted to the research team.

## Results

An overview of the complete validation process is illustrated in Fig 1. Table 1 presents an overview of participating subject experts in step 1 and 2. In total, 22 participants contributed with their expert opinion in the assessment of the intervention's content validity and feasibility, all of which were qualified according to our criteria: patient representatives (n = 11), experienced clinicians of diverse disciplines at specialist and primary care IPRP units (n = 6), and clinicians who were also researchers (n = 5).

### Step 1—Pre-clinical evaluation of content validity

Table 2 shows the I-CVIs and S-CVIs Ave for three assessment rounds. After the final round, the I-CVI ranged from 0.88–1.00 (≥0.78) in most items in all domains, which was interpreted as eVIS attaining "excellent" content validity. The S-CVI/Ave All Items was ≥0.9 from the

**Table 1. Overview of participating subject experts per assessment round and data sources in Step 1 and Step 2.**

| Subject Expert | Profession/ Role | Sex | | Data source | | | | |
|---|---|---|---|---|---|---|---|---|
| | | | Step 1 | Step 2 | | | | |
| | | | Questionnaire, (three assessments) | Physiotherapist-questionnaire (one assessment) | Patient- questionnaire (one assessment) | Diary | Phone/ email support function | Interview |
| 1 | Patient representative | Male | x x x | | | | | |
| 2 | Patient representative | Female | x x x | | | | | |
| 3 | Patient representative | Female | x x - | | | | | |
| 4 | Patient representative | Female | x x x | | | | | |
| 5 | Occupational therapist/ Researcher | Female | x x x | | | | | |
| 6 | Psychologist/ Researcher | Female | x x x | | | | | |
| 7 | Physiotherapist/ Researcher | Female | x x - | | | | | |
| 8 | Occupational therapist/ Researcher | Female | x x x | | | | | |
| 9 | Physiotherapist | Female | x x x | | | | | |
| 10 | Physiotherapist* | Female | x x x | x | | x | x | x |
| 11 | Physiotherapist | Female | | x | | x | x | x |
| 12 | Physiotherapist | Female | | x | | x | x | - |
| 13 | Patient in IPRP | Male | | | x | x | | |
| 14 | Patient in IPRP | Male | | | x | - | x | |
| 15 | Patient in IPRP | Female | | | x | x | | |
| 16 | Patient in IPRP | Male | | | - | x | x | |
| 17 | Patient in IPRP | Female | | | x | x | | |
| 18 | Patient in IPRP | Female | | | x | x | | |
| 19 | Patient in IPRP | Female | | | - | - | | |
| 20 | Physician | Female | | | | | | x |
| 21 | Physician | Male | | | | | | x |
| 22 | Physician/ Researcher | Male | | | | | | x |

* Participated in several steps in the assessment processes

x = assessment participation,— = missing data, IPRP = Interdisciplinary pain rehabilitation program

initial assessment and increased between first and last assessment for all three validity aspects: relevance (0.94 to 0.98), simplicity (0.91 to 0.96), and safety (0.95 to 0.99) (Table 2). The S-CVI/UA also increased over time, with a range of 0.39–0.67 for all validity aspects in the first assessment round, to a range of 0.86–0.95 in the final round ($\geq$ 0.80). In general, ratings for relevance and safety were higher than for simplicity, which highlighted that the dimension simplicity was the most important target for revision. The first round provided 266 comments adjacent to the ratings, round two had 106, and the final round had 59. Topics covered questionnaires and phrasing in PATRON, colour schemes, and personal integrity.

Two items in the Visualization domain had a lower I-CVI, both of which referred to suboptimal simplicity of the timeline graphs in the visualization element. Based on ratings and comments, the graphs joining physical activity (steps/day), pain intensity (1–10), and pain self-efficacy (0–10) were perceived as difficult to read and interpret, hence constituted one of the main targets for revision from the first assessment. Several modifications such as choice of characterizing colours, scaling, and explanatory texts improved ratings of simplicity, however,

**Table 2. Evaluation of content validity in preclinical context, assessment round 1–3; item-level content validity index and summary-content validity index for the validity aspects: Relevance, simplicity, and safety.**

| eVIS domain | Content validity Item | Assessment 1 (n = 10) | | | | | | Assessment 2 (n = 10) | | | | | | Assessment 3 (n = 8) | | | | | |
|---|---|---|---|---|---|---|---|---|---|---|---|---|---|---|---|---|---|---|---|
| | | Relevance | | Simplicity | | Safety | | Relevance | | Simplicity | | Safety | | Relevance | | Simplicity | | Safety | |
| | | Grade | I-CVI | Grade | I-CVI | Grade | I-CVI | Grade | I-CVI | Grade | I-CVI | Grade | I-CVI | Grade | I-CVI | Grade | I-CVI | Grade | I-CVI |
| **Data collection** | | | | | | | | | | | | | | | | | | | |
| Measurement of steps/day | C1 | 4 | 1.00 | 3–4 | 1.00 | 2–4 | 0.90 | 4 | 1.00 | 3–4 | 1.00 | 4 | 1.00 | 4 | 1.00 | 3–4 | 1.00 | 3–4 | 1.00 |
| Choice of assessment domains | C2 | 2–4 | 0.90 | 2–4 | 0.90 | 3–4 | 1.00 | b | | b | | b | | 4 | 1.00 | 4 | 1.00 | 3–4 | 1.00 |
| Data collection PATRON | C3 | 3–4 | 1.00 | 2–4 | 0.80 | 3–4 | 1.00 | 4 | 1.00 | 2–4 | 0.90 | 3–4 | 1.00 | 4 | 1.00 | 3–4 | 1.00 | 3–4 | 1.00 |
| Heading 'Current state' for assessment domains | C4 | 3–4 | 1.00 | 3–4 | 1.00 | 3–4 | 1.00 | b | | b | | b | | 3–4 | 1.00 | 4 | 1.00 | 3–4 | 1.00 |
| NRS 0–10 for pain assessment | C5 | 3–4 | 1.00 | 3–4 | 1.00 | 3–4 | 1.00 | b | | b | | b | | 4 | 1.00 | 4 | 1.00 | 3–4 | 1.00 |
| Question formulation NRS | C6 | 3–4 | 1.00 | 3–4 | 1.00 | a | | b | | b | | b | | 3–4 | 1.00 | 4 | 1.00 | a | |
| PSEQ-2 for self-efficacy assesment | C7 | 2–4 | 0.90 | 1–4 | 0.80 | 1–4 | 0.70 | b | | b | | b | | 3–4 | 1.00 | 3–4 | 1.00 | 3–4 | 1.00 |
| Translation PSEQ-2-introductory text | C8 | 2–4 | 0.60 | 2–4 | 0.80 | a | | 3–4 | 1.00 | 3–4 | 1.00 | a | | 4 | 1.00 | 2–4 | 0.88 | a | |
| Translation PSEQ-2- question 1 | C9 | 2–4 | 0.90 | 3–4 | 1.00 | a | | 3–4 | 1.00 | 3–4 | 0.90 | a | | 3–4 | 1.00 | 3–4 | 1.00 | a | |
| Translation PSEQ-2- question 2 | C10 | 2–4 | 0.90 | 3–4 | 1.00 | a | | 4 | 1.00 | 3–4 | 1.00 | a | | 3–4 | 1.00 | 3–4 | 1.00 | a | |
| Pharmaceutical report function | C11 | 4 | 1.00 | 3–4 | 1.00 | 3–4 | 1.00 | b | | b | | b | 1.00 | 4 | 1.00 | 4 | 1.00 | 4 | 1.00 |
| Registration frequency PATRON | C12 | 3–4 | 1.00 | 3–4 | 1.00 | 3–4 | 1.00 | 3–4 | 1.00 | 4 | 1.00 | 3–4 | 1.00 | 3–4 | 1.00 | 3–4 | 1.00 | 3–4 | 1.00 |
| Registration time PATRON | C13 | 3–4 | 1.00 | 3–4 | 1.00 | 3–4 | 1.00 | b | | b | | b | | 3–4 | 1.00 | 3–4 | 1.00 | 3–4 | 1.00 |
| Overall rating of domain | C14 | 2–4 | 0.90 | 2–4 | 0.80 | 3–4* | 0.90 | 3–4 | 1.00 | 3–4 | 1.00 | 3–4 | 1.00 | 3–4 | 1.00 | 3–4 | 1.00 | 3–4 | 1.00 |
| **S-CVI/Ave Data collection Domain** | | | **0.94** | | **0.90** | | **0.95** | | **1** | | **0.97** | | **1.00** | | **1.00** | | **0.99** | | **1.00** |
| **Visualisation** | | | | | | | | | | | | | | | | | | | |
| Visualisation: 1-day view | C15 | 2–4 | 0.80 | 4 | 1.00 | a | | 4 | 1.00 | 4 | 1.00 | a | | 3–4 | 1.00 | 3–4 | 1.00 | a | |
| Visualisation: 7-day view | C16 | 2–4 | 0.90 | 2–4 | 0.90 | a | | 3–4 | 1.00 | 3–4 | 1.00 | a | | 2–4 | 0.88 | 2–4 | 0.88 | a | |
| Visualisation: 28-day view | C17 | 3–4 | 1.00 | 1–4 | 0.80 | a | | 3–4 | 1.00 | 3–4 | 0.90 | a | | 2–4 | 0.88 | 2–4 | 0.63 | a | |
| Visualisation: Pharmaceuticals | C18 | 2–4 | 0.90 | 3–4 | 1.00 | 3–4 | 1.00 | 3–4 | 1.00 | 3–4 | 1.00 | 3–4 | 1.00 | 3–4 | 1.00 | 3–4 | 1.00 | 3–4 | 1.00 |
| Overall rating of domain | C19 | 2–4 | 0.90 | 2–4 | 0.80 | 3–4 | 1.00 | 3–4 | 1.00 | 3–4 | 1.00 | 3–4 | 1.00 | 2–4 | 0.88 | 2–4 | 0.75 | 2–4 | 0.88 |
| **S-CVI/Ave Visualisation Domain** | | | **0.90** | | **0.90** | | **1.00** | | **1.00** | | **0.98** | | **1.00** | | **0.93** | | **0.85** | | **0.94** |
| **Communication** | | | | | | | | | | | | | | | | | | | |
| eVIS Instruction material | C20 | 2–4 | 0.90 | 2–4 | 0.90 | 3–4 | 1.00 | 3–4 | 1.00 | 3–4 | 1.00 | 3–4 | 1.00 | 3–4 | 1.00 | 3–4 | 1.00 | 3–4 | 1.00 |
| eVIS Support function | C21 | 3–4 | 1.00 | 2–4 | 0.90 | 3–4 | 1.00 | 3–4 | 1.00 | 3–4 | 1.00 | 4 | 1.00 | 3–4 | 1.00 | 2–4 | 0.88 | 3–4 | 1.00 |
| eVIS FAQ-function | C22 | 3–4 | 1.00 | 2–4 | 0.90 | 4 | 1.00 | 3–4 | 1.00 | 3–4 | 1.00 | 4 | 1.00 | 3–4 | 1.00 | 3–4 | 1.00 | 3–4 | 1.00 |
| eVIS usability as means of communication | C23 | 2–4 | 0.90 | 2–4 | 0.80 | 2–4 | 0.90 | b | | b | | b | | 3–4 | 1.00 | 3–4 | 1.00 | 3–4 | 1.00 |
| Overall rating of domain | C24 | 2–4 | 0.90 | 2–4 | 0.80 | 2–4 | 0.90 | 3–4 | 1.00 | 3–4 | 1.00 | 3–4 | 1.00 | 3–4 | 1.00 | 3–4 | 1.00 | 3–4 | 0.88 |
| **S-CVI/Ave Communication Domain** | | | **0.94** | | **0.86** | | **0.96** | | **1.00** | | **1.00** | | **1.00** | | **1.00** | | **1.00** | | **1.00** |
| **eVIS in its entirety** | | | | | | | | | | | | | | | | | | | |

*(Continued)*

**Table 2.** (Continued)

| | | Assessment 1 (n = 10) | | | | | | Assessment 2 (n = 10) | | | | | | Assessment 3 (n = 8) | | | | | |
|---|---|---|---|---|---|---|---|---|---|---|---|---|---|---|---|---|---|---|---|
| eVIS basic idea, purpose, objective | C25 | 3–4 | 1.00 | 3–4* | 0.90 | 3–4* | 0.90 | 4 / 3–4 | 1.00 | 1–4 | 0.90 | 3–4 | 1.00 | 3–4* | 1.00 | 3–4 | 1.00 | 3–4 | 1.00 |
| feasibility/practicability | C26 | 3–4 | 1.00 | 2–4 | 0.80 | 3–4 | 1.00 | 3–4 | 1.00 | 1–4 | 0.90 | 3–4 | 1.00 | 3–4 | 1.00 | 3–4 | 1.00 | 3–4 | 1.00 |
| use/benefit | C27 | 3–4 | 1.00 | 3–4* | 0.90 | 3–4* | 0.90 | 3–4 | 1.00 | 2–4* | 0.80 | 3–4* | 0.90 | 3–4 | 1.00 | 3–4 | 1.00 | 3–4 | 1.00 |
| extent (time and effort) | C28 | 2–4 | 0.90 | 3–4* | 0.90 | 3–4* | 0.90 | 3–4* | 0.90 | 1–4* | 0.80 | 1–4* | 0.80 | 3–4* | 0.88 | 3–4* | 0.88 | 3–4* | 0.88 |
| **S-CVI/Ave eVIS in its entirety Domain** | | | 0.98 | | 0.88 | | 0.93 | | 0.98 | | 0.88 | | 0.93 | | 0.97 | | 0.97 | | 0.97 |
| **S-CVI/Ave All items (1–28)** | | | 0.94 | | 0.91 | | 0.95 | | 1.00 | | 0.96 | | 0.98 | | 0.98 | | 0.96 | | 0.99 |

Domain-specific explications of grades 1 to 4; 1: Not relevant/simple/safe, 2: Needs revision, 3: Relevant/Simple/Safe but needs minor alteration, 4: Very relevant/simple/safe.

a. safety-dimension were not included in question

b. Items omitted in assessment 2

I-CVI (item-level content validity index) = number of experts rating 3 or 4/total number of experts.

* missing, n = 1, Conservative calculation of I-CVI was used (number of experts rating 1 or 2/total number of experts [including the one having a missing answer]).

S-CVI/Ave Domain (scale-level content validity index/average of the domain) = the average I-CVI across items in the domain.

S-CVI/Ave (scale-level content validity index/average) = the average I-CVI across all items in the questionnaire.

Abbreviations: PATRON; Pain and Training Online-web application. NRS 0–10; Numeric Rating Scale. PSEQ-2; 2-Item Short Form Pain Self-Efficacy Questionnaire.

did not reach a satisfactory level in step 1 (see Table 2). This was also reflected in the corresponding S-CVI/Ave Visualization Domain, which ended at 0.85 in the last round. In the Data collection domain, relevance, and simplicity of the proposed translation of the 2-Item Short Form Pain Self-Efficacy Questionnaire (PSEQ-2) was reported as problematic.

The same two topics that emerged as being important for action based on validity ratings were also identified in the commentaries: the visualisation of data in PATRON and the translation of the 2-Item Short Form Pain Self-Efficacy Questionnaire (PSEQ-2). By the end of assessment round 3, although ratings of simplicity of the PSEQ-2 had improved, it became clear that the construct self-efficacy would not be suitable for daily registrations, which led to the item being removed. Instead, a question about pain's interference with daily activities from the West Haven-Yale Multidimensional Pain Inventory (MPI) was added, as this instrument already was translated and validated in Swedish [39]. To enable uniform visualisation of pain intensity and pain interference (in percent) in PATRON, the original MPI scale (0–6) was revised to an 11-point numeric scale (0–10) and the outcome measure was referred to as "interference with daily activities". Clinicians expressed some concerns about the possible negative effects of daily pain assessments that might increase focus on pain and counteract other strategies implemented in IPRPs, which usually aimed to shift focus away from the pain experience. This concern was discussed thoroughly within the consensus group before deciding not to make any alterations, given the importance of validating patient experiences.

## Step 2—Evaluation of content validity and feasibility in a clinical context

After using eVIS in a clinical setting, the post-IPRP assessment conducted by patients (n = 5) and physiotherapists (n = 3) of five content validity items (merged in the feasibility evaluation questionnaires) showed eVIS as valid in the IPRP context. All but three ratings reached I-CVI 1.00 ($\geq$ 0.78) for relevance, simplicity, and safety. One item (C11) emerged noticeably lower. Physiotherapists rated the relevance of the pharmaceutical report function with an I-CVI of 0.33, whereas patients rated the item with an I-CVI of 1.00. Free-text comments revealed physiotherapists had not addressed pharmaceutical consumption during IPRPs, and therefore could not rate the item's relevance appropriately. Physiotherapists rated the safety dimension of the data collection element (C14) as low (I-CVI = 0.33), in contrast to patients (I-CVI = 0.80), however, both commented on an incident in connection to the unintentional linking of Fitbit profiles with notifications from social media. Noticeably, simplicity of the visualization domain (item C19, overall rating of domain) reached I-CVI = 1.0, indicating that simplicity had improved following revisions in step 1.

Participants' feasibility ratings per item are presented in Figs 3 and 4 (patients and physiotherapists, respectively), with ratings $\geq$ 3 indicating satisfactory feasibility. Participants identified eVIS as a feasible intervention within the IPRP context. The main findings are described below with allocated focus areas according to Bowen in brackets and exemplified with patient and physiotherapist quotes from free-text comments in the questionnaires and diaries.

Participants identified a context-based demand for the intervention (*demand*). Four out of five patients and two out of three physiotherapists expressed an intention and willingness to use eVIS in the IPRP context (P-F17 and PT-F23). This was also exemplified from the physiotherapists' rating of patients' interest in the eVIS intervention when presented for them (PT-F2). Patient ratings on item P-F15 and physiotherapists' free text comments, however, indicated potential limitations to participation due to patients being exhausted or lacking time.

> Physiotherapist:.... many (patients) may see it as something stressful, adding yet another thing to do to busy everyday life

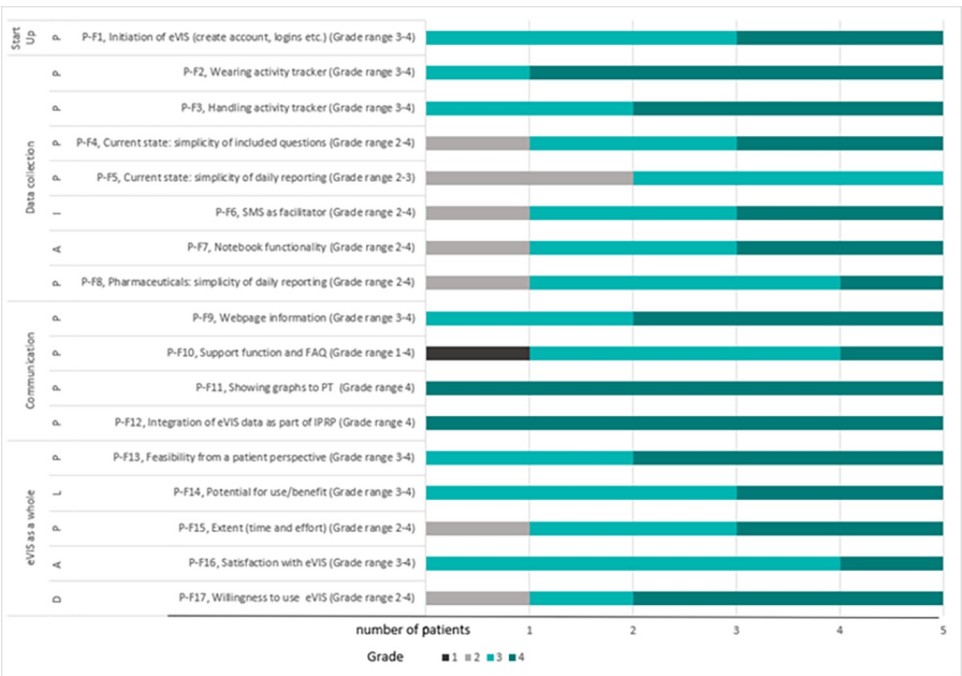

**Fig 3. Evaluation of feasibility in a clinical context.** Ratings post IPRP by patients (n = 5). Domain-specific explications of grades 1: Not at all, 2: To some extent, 3: To a rather large extent, 4: To a large extent. Item P-F17 was rated as extent of agreement with a statement from 1: Strongly disagree to 4: Strongly agree. Ratings ≥ 3 = satisfactory feasibility. Abbreviations: IPRP = Interdisciplinary pain rehabilitation program, P = Practicality, I = Implementation, A = Acceptability, L = Limited Efficacy testing, D = Demand.

Physiotherapists' ratings on items PT-F3 and PT-F11 indicated the intervention was perceived to be useful to patients and satisfactory to both patients and physiotherapists (P-F16 and PT-F22) (*acceptability*). The perceived use/benefit of the intervention was assessed as promising by both patients and physiotherapists, P-F14 and PT-F18 respectively (*limited efficacy testing*).

> *Patient: I think participating in the study has motivated me, even though I haven't been able to identify any direct connections between activity and pain.*

> *Patient: I'm going to get myself one of these watches as it motivates me to move. Movement = less back pain.*

> *Physiotherapist: Graphs provide a good base for communication about pain, mental well-being, ability to be mindfully present, adaptation, and motivation. Better than I expected when I accepted the offer to participate in the study.*

> *Physiotherapist: . . .especially the discussion about this, factors that have impacted (barriers and facilitators) and how to work with these.*

Patients rated the ability to carry out the intervention as feasible (*practicality*) in particular, the SMS-function was considered a facilitator for maintained compliance in data collection of patient-reported outcomes (*implementation*). However, a few participants addressed practicality limitations in the daily report of pharmaceuticals (P-F8), pain intensity, and its interference with daily activities (P-F5) as well as in the communication domain (P-F10), i.e., support function and FAQ (*practicality*).

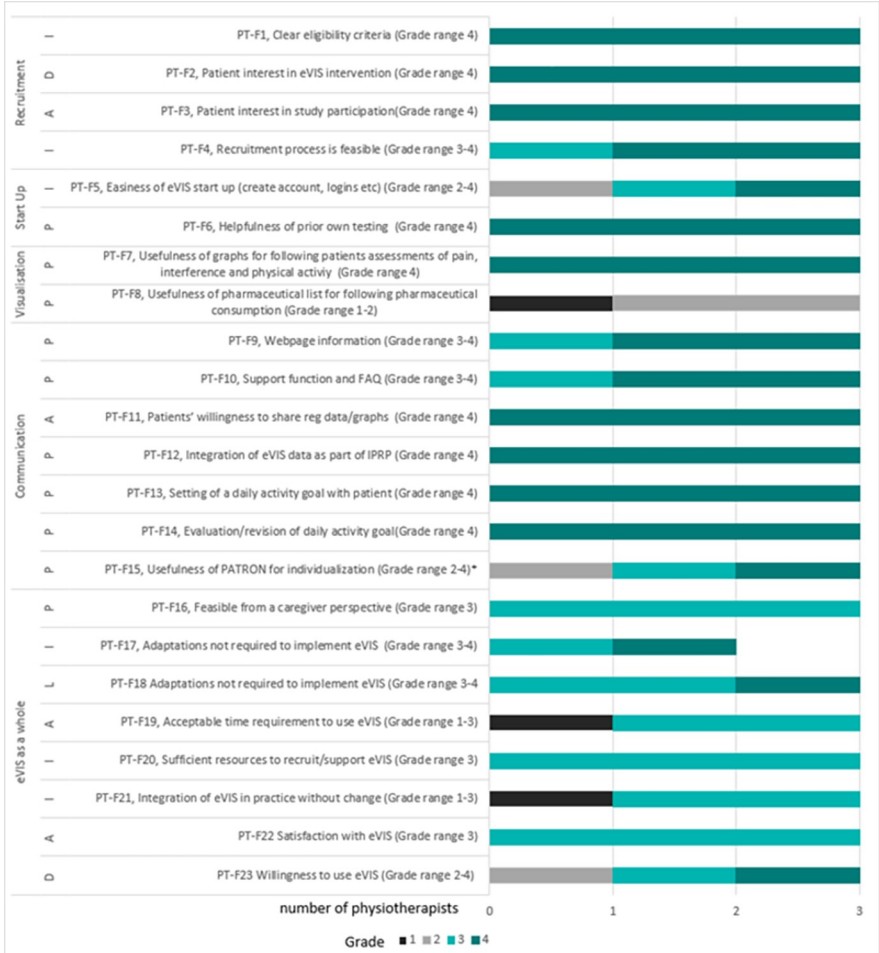

**Fig 4. Evaluation of feasibility in a clinical context.** Ratings post IPRP by physiotherapists (n = 3). Domain-specific explications of grades 1: Not at all, 2: To some extent, 3: To a rather large extent, 4: To a large extent. Item PT-F23 was rated as extent of agreement with a statement from 1: Strongly disagree to 4: Strongly agree. Ratings ≥ 3 = satisfactory feasibility. Abbreviations: IPRP = Interdisciplinary pain rehabilitation program, P = Practicality, I = Implementation, A = Acceptability, L = Limited Efficacy testing, D = Demand. * PT-F15, one physiotherapist follow up was not possible, due to other non-related events.

Physiotherapists' ratings on PT-F5, PT-F21 (*implementation*) reflected factors affecting ease of implementation, as did PT-F19, which reflected perceived difficulties with time requirements and fit within organizational culture (*acceptability*). These results led to follow-up interviews with two of the physiotherapists from step 2 where these items were scrutinized. This led to further development and extended support material for the clinic (checklists and forms were made) to facilitate acceptability and implementation of eVIS, especially regarding recruitment and start up, and time requirements and integration.

Interviews with experts (n = 3) in clinical pharmacological pain management were conducted to evaluate content validity and feasibility of the pharmaceutical report function, which was rated as relevant and safe, but in need of revision to increase simplicity (I-CVI = 1, 1 and 0.67, respectively). This led to the pharmaceutical list in PATRON being substantially extended to encompass all brands pertaining to generic pharmaceuticals, but also to include related drugs for prevalent comorbidities associated with chronic pain, such as anxiety and depression disorders. In addition, the functionality of handling the pharmaceutical list was transferred to the research team instead of the software team, to simplify the making of amendments.

A final aspect of feasibility was the evaluation of the data export function from PATRON, which established that the prerequisites for export of synchronized data from the activity tracker and patient registrations were satisfactory and valid for data analyses in a forthcoming registry-based randomized controlled trial.

## Discussion

### Principal findings

In this study, relevance, simplicity, safety, acceptability, practicality, implementation, demand, and limited efficacy testing of eVIS was evaluated by subject experts, and iteratively improved upon by consensus panel, in a preclinical and clinical context. The relevance, simplicity, and safety of eVIS emerged with initial acceptable levels, which improved further during the iterative rounds and development process. The intervention's acceptability, practicality, implementation, demand, and limited efficacy emerged as satisfactory when assessed in practice within the IPRP context. Areas of development were primarily the visualisation domain, safety features, the pharmacological report function, and recruitment processes. Identified demerits led to continuous protocol revisions, as well as immersed data collection by interviews with experts in certain areas. The findings establish the eVIS intervention's content validity and feasibility.

### Methodological considerations

**Step 1—Pre-clinical evaluation of content validity.** According to Polit & Beck (2007) as well as Grant (1997), the soundness of the interrater estimate in content validity is strongly influenced by the selection of experts [32, 41]. We therefore followed recommendations to assure relevant subject expertise was included by recruiting subjects from essential fields relating to the rehabilitation of chronic pain and intervention development. In addition, to adjust the risk of inflated estimates, we ascertained that the study incorporated a sufficient number of experts [41, 44]. Despite such efforts, we identified a lack of input on the pharmaceutical report function of eVIS. This was remediated through video interviews with specifically recruited subject experts, resulting in a deepened evaluation and refinement on the pharmaceutical report function.

Following recommendations, [31, 32, 43] we collapsed the four response options into two categories for our CVI calculation. It has been argued that less detail in information is derived from the interrater estimate [44]. However, in this study, expert ratings were to a large degree coherent. Moreover, valuable addition from rich free-text comments provided enough detail to guide the iterative development and evaluation process.

**Step 2—Evaluation of content validity and feasibility in a clinical context.** To our knowledge, methodological guidelines on how to conduct feasibility evaluations are scarce, despite strong recommendations to conduct such studies. We evaluated vital parts of the intervention, rated it after an actual test period, and categorized our feasibility items in terms of five relevant focus areas [29]. We used robust methodology to ensure consensus in the categorization of included items in both the patient and physiotherapist questionnaires. Still, these may be open to interpretation, especially since some items are related to different focus areas depending on whose perspective they represent.

We used diverse data sources (questionnaires, diaries, interviews, and support errands) to increase input on eVIS feasibility. The sample providing assessments on feasibility was rather small, but as the evaluation constituted part of a larger evaluation process, the basis was deemed to be sufficient. Only three physiotherapists participated, however, their ratings were based on their accumulated experience of introducing eVIS to multiple patients, which gave

them a broader understanding of the intervention's applicability to a heterogenous population. To minimize the potential loss of information, we synthesized feasibility ratings descriptively.

## Discussion of findings

eVIS was systematically developed according to the Medical Research Council's updated framework for the development and evaluation of complex interventions [35]. Stakeholders (patients, clinicians, researchers) were actively involved throughout the development, testing and evaluation stages. We hypothesized that eVIS would be a valid and feasible intervention. We evaluated content validity 'in theory' in a pre-clinical context, and 'in praxis' in a clinical context, where feasibility was also evaluated. The evaluations emerged with favourable results, and in support of our hypothesis.

**Content validity of eVIS.** We found consensus already from the first assessment that eVIS to the main part was considered valid, based on item level indicators of relevance, simplicity and safety, and with only a few items below our cut off. Patients, caregivers, and researchers were generally positive towards the features on which eVIS was based.

Regarding the *relevance* of eVIS, the key uncertainties and focus of attention was on the included self-reported measures. We found that the construct of self-efficacy was not conceptually relevant for daily measuring. Instead, a more tangible question about perceived interference with daily activities, [39] replaced the initial PSEQ-2, which also had the benefit of being validated in Swedish. Another aspect related to the self-reported measures was the daily assessment of pain intensity. It was indicated by some clinicians as possibly counteractive to the IPRP concepts, as frequently focusing on pain assessment might increase patients' perceived pain intensity. We found no evidence in support of reactive effects from daily pain assessment. Instead, repeated reporting of pain reportedly improves measurement reliability [45]. Moreover, evidence suggests the reflective act of recording data may result in increased understanding of the variability of pain, as well as recognition of time with less pain, and the observed relationship between pain intensity and activity, [46] which in itself can be perceived as a validation of patients' experience. Moreover, it is reasonable to believe that pain intensity to some degree will co-vary with activity and interference and therefore is of importance to study, in order to increase our understanding of how these factors interact.

*Simplicity* was the part that needed most attention, based on our results. To increase usability and satisfaction with the system, we carried out extensive revisions based on repeated, formative assessments. A strength in this process was the help we received from relevant experts, not only from the point of view of subject experts but also from the field of audio-visual communication, giving guidance, for instance on avoiding unintentional signalling through our choice of colours. Altogether, and in dialogue with the software developers, the interface improved markedly.

*Safety* emerged with high ratings in almost all areas, however, during the clinical testing a safety breach occurred which resulted in lower ratings and comments from both patients and physiotherapists. Although we did not anticipate any major safety issues, safety was an important aspect to validate. In conjunction with relevance and simplicity, commonly used for evaluation in questionnaire development, safety aspects play an important role in intervention development. From previous experiences of validating a safety-critical intervention [33] we were certain that the eVIS intervention would also benefit from systematic validation from subject expertise to identify possible overlooked risks and increase safety confidence. Contrary to many other eHealth applications, eVIS and PATRON do not include specific treatment advice or exercise modules, as this is handled within the clinical setting. Hence, the important parts of our validation process were related to patient integrity. Following discussions on the

question "who has access to my daily data?" reinforced our intention to only enable clinicians to see data when patients themselves logged in and showed them, instead of clinicians having access to monitor or "survey" from a distance. Moreover, the incidence during step 2 evaluation resulted in important actions being taken to prevent re-occurrence.

**Feasibility of eVIS.** Results from our clinical evaluation of eVIS were in line with other recent feasibility evaluations of digital tools designed to support patients with chronic pain and chronic diseases [47–49]. Outcomes such as system use, usability, acceptability, and ease of use are emphasized as important factors in the achievement of high adherence to digital self-management tools [48]. In our study, stakeholders assessed eVIS as in demand, acceptable, practical, and possible to implement within IPRP, which are important factors to maintain a high adherence to the intervention over time [47, 50]. Moreover, Ricciardo & Pandya (2020) concluded that the perceived ease of use and features facilitating knowledge acquisition were key features of an eHealth intervention designed to facilitate patient self-management [49]. In an evaluation of patients with chronic diseases and their 'needs and requirements when using eHealth and self-management, individual tailoring of the tool are addressed as paramount [48]. In 2018, Karlsson et al. published a report on how patients living with chronic pain experience physical activity and exercise [50]. In the report, it was confirmed that many patients living with chronic pain experience significant barriers to performing physical activity. Factors such as motivation, self-efficacy, action control (i.e., transferring intention into action to accomplish a certain behaviour), interactions, and profuse contact with healthcare providers are emphasized as necessary requirements that need to be addressed in the process of the individualization of physical activity level.

Patients and physiotherapists rated the limited-efficacy test as high, suggesting a clinical demand for a tool such as eVIS. Contrary to many other eHealth interventions, eVIS is not intended for patients to use on their own and it does not provide any individualized advice or exercises, rather, it has been developed for use in the IPRP context. We believe the IPRP setting, with physiotherapists and other experts from the interdisciplinary team, e.g., occupational therapists, psychologists, or physicians, are an essential part of the successful and effective use of eVIS. For patients with chronic and complex pain conditions, to set and adapt an individualised physical activity goal necessitates expertise and collaboration with the patient, taking into account personal resources, intentions, and available time plan. It appears eVIS may be a useful tool in bridging IPRP clinical context with treatment individualisation and self-management.

## Strengths and limitations

Finally, ascertaining an intervention's content validity and feasibility before proceeding to an effectiveness trial is considered as a significant strength and is highly recommended [25–29, 33]. To date, the development, test, and evaluation stages of the intervention have taken approximately 1.5–2 years to complete. Although documenting the content validity and feasibility of an intervention may seem expensive in terms of time and human resources, its importance warrants these costs.

The study is not without limitations. In retrospect, it would have been of value to use a structured protocol for more transparency of the consensus process and following decisions. Moreover, our findings are based on a sample of subject expertise and may therefore not apply to the whole target population. However, the study provided enough detail to confidently set the intervention protocol and to proceed to the next step of trial. In accordance with the updated framework for the development and evaluation of complex interventions, the follow-on development and evaluation of the eVIS intervention will be performed through a

randomized pilot study before proceeding on to an effectiveness trial [20]. The pilot study will enable a further and more robust evaluation of key feasibility outcomes to be made, and on a larger sample.

## Conclusions

We conclude that the proposed domains and features of the eVIS intervention are deemed valid in its content and feasible in the IPRP context. The consecutive step-by-step evaluation process enabled careful intervention development with revisions made in close collaboration with stakeholders. Findings implicate a robust base ahead of the forthcoming effectiveness trial.

## Acknowledgments

The authors would like to acknowledge Associate Professor Thorbjörn Swenberg, Dalarna University and Nordforce Technology AB, Stockholm for their valuable contributions in the development and validation process.

## Author Contributions

**Conceptualization:** Mathilda Björk, Björn O. Äng, Linda Vixner.

**Data curation:** Elena Tseli.

**Formal analysis:** Elena Tseli, Veronica Sjöberg, Mathilda Björk, Björn O. Äng, Linda Vixner.

**Funding acquisition:** Mathilda Björk, Björn O. Äng, Linda Vixner.

**Investigation:** Elena Tseli, Veronica Sjöberg.

**Methodology:** Elena Tseli, Mathilda Björk, Björn O. Äng, Linda Vixner.

**Project administration:** Elena Tseli, Veronica Sjöberg, Linda Vixner.

**Resources:** Linda Vixner.

**Supervision:** Linda Vixner.

**Validation:** Veronica Sjöberg, Linda Vixner.

**Writing – original draft:** Elena Tseli, Veronica Sjöberg.

**Writing – review & editing:** Elena Tseli, Veronica Sjöberg, Mathilda Björk, Björn O. Äng, Linda Vixner.

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
