## [Decision Letter · Decision Letter 0]

20 Sep 2022

PONE-D-22-23979Evaluation of content validity and feasibility of the eVISualisation of physical activity and pain (eVIS) intervention for patients with chronic pain participating in interdisciplinary pain rehabilitation programsPLOS ONE

Dear Dr. Tseli,

Thank you for submitting your manuscript to PLOS ONE. After careful consideration, we feel that it has merit but does not fully meet PLOS ONE’s publication criteria as it currently stands. Therefore, we invite you to submit a revised version of the manuscript that addresses the points raised during the review process.

We look forward to receiving your revised manuscript.

Kind regards,

Alejandro Vega-Muñoz, Ph.D.

Academic Editor

PLOS ONE

2. We noted in your submission details that a portion of your manuscript may have been presented or published elsewhere. No, except from a recent poster presentation at the

Conference: 12th Congress of the European Pain Federation EFIC, Dublin, Ireland, 27-30 April 2022, entitled: "Content Validity and Feasibility of the eVISualisation of physical activity and pain (eVIS) intervention in Interdisciplinary Pain Rehabilitation Programs: Valuable steps in a systematical development and evaluation process". Figures and data are not identical with our submitted manuscript to PLOS. Please clarify whether this conference proceeding was peer-reviewed and formally published. If this work was previously peer-reviewed and published, in the cover letter please provide the reason that this work does not constitute dual publication and should be included in the current manuscript.

Additional Editor Comments :

Dear Author, based on the comments of both reviewers my decision is that your work requires a major revision.

Please consider the comments of both reviewers, to elaborate your response and submit an improved version of your manuscript.

Reviewers' comments:

Reviewer's Responses to Questions

**Comments to the Author**

1. Is the manuscript technically sound, and do the data support the conclusions?

Reviewer #1: Yes

Reviewer #2: Partly

2. Has the statistical analysis been performed appropriately and rigorously? 

Reviewer #1: I Don't Know

Reviewer #2: I Don't Know

3. Have the authors made all data underlying the findings in their manuscript fully available?

Reviewer #1: No

Reviewer #2: Yes

4. Is the manuscript presented in an intelligible fashion and written in standard English?

Reviewer #1: Yes

Reviewer #2: Yes

5. Review Comments to the Author

Reviewer #1: Dear editors and researchers,

I would like to thank you for the opportunity to review this manuscript, as well as congratulate the researchers for their work and thank you for your intent to be published.

We have an article that aims to assess the validity and reliability of the eVIS intervention in chronic pain patients participating in interdisciplinary pain rehabilitation programmes. Research methodologies that include quantitative and qualitative aspects are welcome, as they provide very interesting information, despite their complexity.

The authors have done a good job of introducing the subject matter, contextualising and grounding the research, as well as describing the processes involved in the research.

I have no major problems with this work, only minor recommendations and comments.

The content of the abstract is adequate, making a brief presentation of the introduction, summarising the most important aspects of the method, highlighting the most relevant results and conclusions. However, I would recommend the authors not to include concepts used in the title and/or abstract in the keywords, thus increasing the chances of being listed in search engines.

Indeed, chronic musculoskeletal pain is a complex disease, often used generically when the exact causes of the pain are not known, encompassing many types of pain, so its definition is not simple. The authors could make some reference to the current definition of chronic musculoskeletal pain, its possible causes and whether the origin could modify the intensities and effects. The rest of the introduction is well directed towards the objectives of the research, although it would be advisable to include what the hypothesis was.

The methodology is well described, but I have some doubts regarding the selection of participants, although I do not feel competent to assess whether the number of participants and their selection was adequate.

The results section is clearly developed, with sufficient information on all findings, including tables that facilitate the understanding of the quantitative data, as well as explanations of the qualitative aspects.

The discussion presents the main findings and debate on the encounters with classical and current references.

Best regards

Reviewer #2: Thank you for the opportunity to review this manuscript. I would like to congratulate the authors on their work and for their contribution to the scientific literature. Their work is important and relevant to optimize the health of patient living with chronic pain. However, I have highlighted several aspects that need clarifications, along with comments and major preoccupations concerning their paper, which I feel the authors should consider.

Introduction:

- The problematic is well described, and its relevance is well put forward by the authors

- The rationale for the proposed intervention is clear. A brief description is present which, in my opinion, lacks details (i.e., mechanisms of action). Yet, it could still be considered adequate for the introduction section.

- A theoretical framework is highlighted.

Objectives:

- We believe that the formulation of the objectives is misleading and could be improved. After reading the methods section, there is clearly more than 1 objective contained in the paper, but the formulation does not reflect that. Thus, it should be divided into 2 (i.e., content validity and feasibility) or even 3 distinct objectives since the methods for each objective vary greatly. This issue has great repercussion in the readability of the overall paper and particularly the methods section (see comments below).

Methods:

- For better readability, we invite the authors to consider fragmenting their methods by objective (i.e., step 1 and 2) as each objective is attained with a specific method.

- Table 1 provides very little information on the participants. Were other characteristics of the sample collected other than sex? We suggest to the authors to provide additional information on the participants of the study if they have such data. Otherwise, revise (major) Table 1. Moreover, information regarding participants represents a result, which is usually presented in the results section.

- Recruitment of the study participants for step 1 lacks details. How were the subject expert recruited? (line 146) How was “relevant knowledge” defined? (line 141). The authors need to make it explicit – how were the potential participants deemed/considered as experts? What were the criteria?

- Recruitment process needs further description. Currently, the risk of selection bias seems highly likely based on the details contained in the paper.

-

- Same issue related to the recruitment process of the study participants is present for step 2.

- Why only 2 physiotherapists were used as subject experts for follow-up interview? (line 177-8)? Another selection bias? Only 2 PT is enough to provide solid data for this objective?

- The intervention is well described. However, it would be relevant to have more information on its mechanism of action, and more specifically on the communication part. How is the information gathered via the eVIS intervention used by the health professionals who work within the IPRP context? Also, it would be interesting to provide additional information on how the documentation and visualisation parts of the eVIS intervention translate into the communication part.

- As for the content Validity Index (CVI), there is no specific link to the objective and how it was operationalized

- The assessment rounds for the consensus process in step 1 is hard to follow. It lacks details on how the consensus experts worked together and which changes were applied based on the consecutive rounds. What were the criteria to make such changes? Was the consensus process standardized in some way? How were the information analyzed by the subject experts?

- The assessment of the qualitative data from the questionnaires (open question), interviews and journaling are poorly described. No mentions of specific data analysis methods (i.e., coding, multiple evaluators, themes, software used?). No mentions of an analysis framework.

- Why were there 3 rounds of assessment for the content validity? This should be made explicit

- Overall, the method section is hard to follow based on multiple methods for multiples objectives.

Results:

- There is no information on the characteristics of the participants in either step 1 or 2.

- We believe the sample of n=8 (5 patients and 3 physiotherapists) is very small to properly assess feasibility. This should be addressed (explained) or at least mentioned as a limit of the study.

- Table 2, in its current form, is very hard to interpret for the reader. Maybe too much information is presented…

Discussion:

- The discussion is well written and supported by the literature. However, the interpretation of the results by the authors should be much more nuanced considering the limits of the study. We urge the authors to add a proper section on the limits and strength of the study, including sample size

Based on these observations, my opinion is to recommend major revisions of the manuscript. The major flaws include the poor congruence between the objectives and the described methodology, which is hard to follow. We suggest to the authors to divide more clearly the different objectives pursued in the article. Also, we believe that there is important information lacking in the description of the methodology concerning aspects such as the recruitment process of the participants, definition of subject experts, the lack of transparency of the consensus process, the data analysis of qualitative data and a more nuanced interpretation of the results based on the limits highlighted. In its current form, there are major bias that could be addressed by a more structured and detailed methodology. Moreover, we are wondering if the proposed article is too big in its current form and whether it should be preferable to focus on content validity and feasibility in 2 separate articles.

6. PLOS authors have the option to publish the peer review history of their article (what does this mean?). If published, this will include your full peer review and any attached files.

Reviewer #1: No

Reviewer #2: No

---

## [Author Response · Author response to Decision Letter 0]

18 Nov 2022

Editor’s and Reviewer Comments - point-by-point response.

Please note that indicated pages and rows, in the column Action, refer to the revised paper without tracked changes, the file labeled 'Manuscript'.

Editor’s Comments 

Author response 

Thank you. We have checked the PLOS ONE’s style requirements once more and believe we now have formatted correctly. Revision of file naming. 

2. We noted in your submission details that a portion of your manuscript may have been presented or published elsewhere. “No, except from a recent poster presentation at the Conference: 12th Congress of the European Pain Federation EFIC, Dublin, Ireland, 27-30 April 2022, entitled: "Content Validity and Feasibility of the eVISualisation of physical activity and pain (eVIS) intervention in Interdisciplinary Pain Rehabilitation Programs: Valuable steps in a systematical development and evaluation process". Figures and data are not identical with our submitted manuscript to PLOS.” 

Please clarify whether this conference proceeding was peer-reviewed and formally published. If this work was previously peer-reviewed and published, in the cover letter please provide the reason that this work does not constitute dual publication and should be included in the current manuscript. 

Author response

To clarify, the submitted abstract was peer-reviewed by the Conference’s Scientific Programme Committee but not formally published. The poster, which was neither peer-reviewed nor formally published, depicted the study process and conclusion but includes no quantitative or qualitative data. This conference proceeding does not constitute dual publication of the paper. 

This is now commented in Cover letter. 

During our scrutiny of this query we realized that the concept figure of eVIS was previously published in our study protocol - and therefore we have now attributed Figure 2 to the source reference and CC license, CC BY-NC 4.0. 

3. We note that you have indicated that data from this study are available upon request. PLOS only allows data to be available upon request if there are legal or ethical restrictions on sharing data publicly. For more information on unacceptable data access restrictions, please see http://journals.plos.org/plosone/s/data-availability#loc-unacceptable-dataaccess-restrictions.

a) If there are ethical or legal restrictions on sharing a de-identified data set, please explain them in detail (e.g., data contain potentially sensitive information, data are owned by a third-party organization, etc.) and who has imposed them (e.g., an ethics committee).

Please also provide contact information for a data access committee, ethics committee, or other institutional body to which data requests may be sent.

Author response

Thank you for pointing out this ambiguity in our statement. 

Our reply to the Data availability query has now been changed to only a 'No' and we have revised the Data Availability Statement for better clarity on the ethical restrictions imposed by the ethics committee and legal restrictions. Moreover, we added contact information to institutional body, complementary to principal investigator.

Action: 

-Revision of Data Availability Statement.

-Comment in Cover letter.

Revised Data Availability Statement: 

Data cannot be shared publicly because of ethical restrictions involving human research participant information.

Data contains potentially identifying information and sensitive patient information. Data are collected from a small group of participants and especially qualitative data, such as free-text comments in questionnaires and diary notes, risk compromise participant privacy.

Data is only available from the Dalarna University Institutional Data Access for researchers who meet the criteria for access to confidential data, after approval from the principal investigator. Data requests may be sent to the Dalarna University Institutional Data Access, dataskydd@du.se. 

De-identified individual data (participant ratings) used in this publication will be made available on reasonable request to the principal investigator. However, such a request must lie within the limits of the ethical approval for this project. In addition, to uphold data safety and other legal aspects, relevant agreements must be established prior to data being available outside the research group.

Reviewer #1: Author Response

The content of the abstract is adequate, making a brief presentation of the introduction, summarising the most important aspects of the method, highlighting the most relevant results and conclusions. However, I would recommend the authors not to include concepts used in the title and/or abstract in the keywords, thus increasing the chances of being listed in search engines

Author response

Thank you for pointing this out. We have now rephrased three of our key words to increase our visibility in search engines; Content validity and Feasibility have been replaced by Evaluation study and Physical activity with Exercise. Page 3, rows 48-49

Indeed, chronic musculoskeletal pain is a complex disease, often used generically when the exact causes of the pain are not known, encompassing many types of pain, so its definition is not simple. The authors could make some reference to the current definition of chronic musculoskeletal pain, its possible causes and whether the origin could modify the intensities and effects. 

Author response

Thank you for this recommendation. 

In the Introduction, we have now chosen to provide some clearer distinctions between chronic pain, and the recent advances in classifications relating to these. Page 4, row 54-60

The rest of the introduction is well directed towards the objectives of the research, although it would be advisable to include what the hypothesis was.

Author response

Thank you for this suggestion. We have now elaborated on our hypothesis in the Introduction and Discussion. Page 6, row 115-116 and Page 32, row 615-618

The methodology is well described, but I have some doubts regarding the selection of participants, although I do not feel competent to assess whether the number of participants and their selection was adequate.

Author response

Thank you. We have now revised the methods section regarding participant selection and also commented on this query in the discussion. Page 9-11, Page 30, row 572-584, Page 31, row 604-609, Page 35, row 701-708 

The results section is clearly developed, with sufficient information on all findings, including tables that facilitate the understanding of the quantitative data, as well as explanations of the qualitative aspects.

Author response

Thank you

The discussion presents the main findings and debate on the encounters with classical and current references.

Author response

Thank you

Reviewer #2: Author Response

Thank you for the opportunity to review this manuscript. I would like to congratulate the authors on their work and for their contribution to the scientific literature. Their work is important and relevant to optimize the health of patient living with chronic pain. However, I have highlighted several aspects that need clarifications, along with comments and major preoccupations concerning their paper, which I feel the authors should consider.

Author response 

Thank you

Introduction:

- The problematic is well described, and its relevance is well put forward by the authors

- The rationale for the proposed intervention is clear. A brief description is present which, in my opinion, lacks details (i.e., mechanisms of action). Yet, it could still be considered adequate for the introduction section.

- A theoretical framework is highlighted.

Author response

Thank you. We have now built on the rationale a little earlier in the text, and condensed the descriptions of behavioral change techniques, with referral to theory on proposed mechanisms of action. Page 4, row 72-73 and Page 5, row 81-87 

Objectives:

- We believe that the formulation of the objectives is misleading and could be improved.

After reading the methods section, there is clearly more than 1 objective contained in the paper, but the formulation does not reflect that. Thus, it should be divided into 2 (i.e., content validity and feasibility) or even 3 distinct objectives since the methods for each objective vary greatly. This issue has great repercussion in the readability of the overall paper and particularly the methods section (see comments below).

Author response

We have now made revisions according to your suggestion. We have rephrased the objectives and divided them into two: 

1) To evaluate the content validity of eVIS in a pre-clinical context

2) To evaluate the content validity and feasibility of eVIS in a clinical context

Page 6, Row 120-123.

Methods:

- For better readability, we invite the authors to consider fragmenting their methods by objective (i.e., step 1 and 2) as each objective is attained with a specific method.

Author response

Thank you for this suggestion.

We have now restructured the Methods section and fragmented our methods by objective, indicated by Step 1 and Step 2 headings, in accordance with our objectives. Steps 1 and 2 are further detailed under Study design and context. 

As a consequence, we also moved the description of the eVIS intervention up in Methods, under the heading Subject of evaluation, before the detailing of the two methods.

Page 6 to 18: -Consequent headings in accordance with two objectives. 

-Page 7, 135-141

-Transfer of the description of the eVIS intervention, from page 10 to 8.

-Revision Fig 1

- Table 1 provides very little information on the participants. Were other characteristics of the sample collected other than sex? We suggest to the authors to provide additional information on the participants of the study if they have such data. 

Otherwise, revise (major) Table 1. 

Moreover, information regarding participants represents a result, which is usually presented in the results section.

Author response

Thank you for pointing this out. 

In Step 1 and the questionnaire used for Assessment 1, we included a few questions on characteristics to which participants were prompted depending on participant category (patient, clinician, or researcher). 

These were mainly intended to establish participants’ relevant topic expertise, that they were representative samples of the content domain. We did not collect any additional information. 

We have now provided some more information on the participants and reordered text to give a more coherent description of their expertise, as part of our methods section. Page 9-10, 194-236

We also moved Table 1 to Results section, for an overview of included subjects in the whole study. Page 18-19, Table 1 

- Recruitment of the study participants for step 1 lacks details. How were the subject expert recruited? (line 146) How was “relevant knowledge” defined? (line 141). The authors need to make it explicit – how were the potential participants deemed/considered as experts?

What were the criteria?

Author response

Thank you, we agree the eligibility criteria might be perceived as too vague. We have now elaborated more on our descriptions of what constitutes subject expertise, and how we approached the recruitment process to attain suitable participants on the basis of their experience of the phenomenon of interest.

A strategic recruitment of study participants was applied in step 1 to achieve a broad representation of the eVIS target population, combined with documented expertise either clinical or research-based within the fields relating to eHealth, physical activity, behavior change, outcome measures etc. The expertise area was verified by the initial questions in the questionnaire used for Assessment 1. 

From our recruitment process, we were confident that subjects were representative samples of the content domain. Page 9-11, row 184-236

- Recruitment process needs further description. Currently, the risk of selection bias seems highly likely based on the details contained in the paper. 

Author response

We have now elaborated more on how the recruitment was done, by adjoining text sections into more coherent parts and complemented with details. Page 9-10, row 184-211

- Same issue related to the recruitment process of the study participants is present for step 2. 

Author response

We have now described the recruitment process in step 2 in more detail. Page 10-11, row 214-232

- Why only 2 physiotherapists were used as subject experts for follow-up interview? (line 177-8)? Another selection bias? Only 2 PT is enough to provide solid data for this objective? 

Author response

All three physiotherapists were invited to the follow-up interview. However, one was unable to participate, due of staff restrictions due to the pandemic. We have now added the reason for one missing in the text. 

As our follow-up interview mainly concerned feasibility aspects of the Start-up process, on how to best accommodate needs from the clinic to make initiation and patient recruitment as easy as possible, we believe the discussions provided enough input for our following revisions. Page 11, row 227-228

- The intervention is well described. 

However, it would be relevant to have more information on its mechanism of action, and more specifically on the communication part. 

How is the information gathered via the eVIS intervention used by the health professionals who work within the IPRP context? 

Also, it would be interesting to provide additional information on how the documentation and visualisation parts of the eVIS intervention translate into the communication part.

Author response

We briefly address the potential mechanisms of action in the Introduction section. These were not explicitly in the scope of this study.

The information gathered via the eVIS intervention is intended to be communicated between caregivers and patients in in synchronous one to one meeting, that is, the patient logs in and shows the documentation. 

There is no formalised directive on how this communication is to be implemented, it is up to the caregiver and patient to use and integrate the clinical tool as fits them within the ordinary care.

1. Page 5

2. Page 8, row 174

3. Page 9, row 177-178

- As for the content Validity Index (CVI), there is no specific link to the objective and how it was operationalized

Author response

Thank you, we understand the rationale for using the CVI as indicator of content validity was missing. We have now tried to clarify how the CVI fits into the process of judgment quantification by adding text in two different places in the manuscript, under ‘Study design and context’ and ‘Outcomes’. 

Page 7, row 128-134

Page 12, row 257-259

Page 12-13, row 273-275

- The assessment rounds for the consensus process in step 1 is hard to follow. It lacks details on how the consensus experts worked together and which changes were applied based on the consecutive rounds. 

What were the criteria to make such changes? 

Was the consensus process standardized in some way? 

How were the information analyzed by the subject experts?

Author response

Thank you. An introduction on the consensus panel task has now been added under ‘Study design and context’ as well as a more detailed explication of the consensus process in step 1, under ‘Data collection and assessment processes’.

Page 7, row 131-134

Page 11-12, 239-253

Page 14, row 315-320

- The assessment of the qualitative data from the questionnaires (open question), interviews and journaling are poorly described. No mentions of specific data analysis methods (i.e., coding, multiple evaluators, themes, software used?). No mentions of an analysis framework.

Author response

We have now clarified that no specific qualitative analysis framework/ methodology was applied in the interpretation of the free text comments in the questionnaire, diaries, etc .

Page 17, row 372-374

- Why were there 3 rounds of assessment for the content validity? This should be made explicit

Author response

Thank you, we have now explained the reason for the 3 rounds. Page 14, row 299-301

- Overall, the method section is hard to follow based on multiple methods for multiples objectives.

Author response

The method section is restructured, and headings have been revised in accordance with the two steps/ objectives. The titles for Figures 3 and 4 are rephrased, to be in line with objective 2. We have also revised Figure 1 accordingly. Page 6-18 and Figure 1

Results:

- There is no information on the characteristics of the participants in either step 1 or 2.

Author response

See note above about transfer of Table 1.Page 18, row 406-411.

- We believe the sample of n=8 (5 patients and 3 physiotherapists) is very small to properly assess feasibility. This should be addressed (explained) or at least mentioned as a limit of the study.

Author response

Thank you, we agree this should be commented on. It was mentioned in the Methodological considerations before, but we have now addressed this matter in more detail in the Limitations section. Page 31, row 604-609 and Page 35, row 699-708

- Table 2, in its current form, is very hard to interpret for the reader. Maybe too much information is presented…

Author response

Thank you for pointing this out. We have now added more descriptive text to the Table 2 title and re-formatted the table columns to make reading more intuitive for the reader. Page 22-24; Table 2- revised

Discussion:

- The discussion is well written and supported by the literature. However, the interpretation of the results by the authors should be much more nuanced considering the limits of the study. We urge the authors to add a proper section on the limits and strength of the study, including sample size

Author response

We have now added a ‘Strengths and limitations’ section. Page 35, row 692-708

Based on these observations, my opinion is to recommend major revisions of the manuscript. The major flaws include the poor congruence between the objectives and the described methodology, which is hard to follow. We suggest to the authors to divide more clearly the different objectives pursued in the article. Also, we believe that there is important information lacking in the description of the methodology concerning aspects such as the recruitment process of the participants, definition of subject experts, the lack of transparency of the consensus process, the data analysis of qualitative data and a more nuanced interpretation of the results based on the limits highlighted. In its current form, there are major bias that could be addressed by a more structured and detailed methodology. Moreover, we are wondering if the proposed article is too big in its current form and whether it should be preferable to focus on content validity and feasibility in 2 separate articles.

Author response

Thank you for pointing at these important gaps in our manuscript - we have now re-structured and detailed more on the methodology and we have also addressed the study limitations.

---

## [Decision Letter · Decision Letter 1]

31 Jan 2023

PONE-D-22-23979R1Evaluation of content validity and feasibility of the eVISualisation of physical activity and pain (eVIS) intervention for patients with chronic pain participating in interdisciplinary pain rehabilitation programsPLOS ONE

Dear Dr. Tseli,

Thank you for submitting your manuscript to PLOS ONE. After careful consideration, we feel that it has merit but does not fully meet PLOS ONE’s publication criteria as it currently stands. Therefore, we invite you to submit a revised version of the manuscript that addresses the points raised during the review process.

We look forward to receiving your revised manuscript.

Kind regards,

Alejandro Vega-Muñoz, Ph.D.

Academic Editor

PLOS ONE

Journal Requirements:

Reviewers' comments:

Reviewer's Responses to Questions

**Comments to the Author**

1. If the authors have adequately addressed your comments raised in a previous round of review and you feel that this manuscript is now acceptable for publication, you may indicate that here to bypass the “Comments to the Author” section, enter your conflict of interest statement in the “Confidential to Editor” section, and submit your "Accept" recommendation.

Reviewer #1: All comments have been addressed

Reviewer #3: (No Response)

2. Is the manuscript technically sound, and do the data support the conclusions?

Reviewer #1: Yes

Reviewer #3: Yes

3. Has the statistical analysis been performed appropriately and rigorously? 

Reviewer #1: Yes

Reviewer #3: Yes

4. Have the authors made all data underlying the findings in their manuscript fully available?

Reviewer #1: Yes

Reviewer #3: Yes

5. Is the manuscript presented in an intelligible fashion and written in standard English?

Reviewer #1: Yes

Reviewer #3: Yes

6. Review Comments to the Author

Reviewer #1: Dear Authors,

I have no further considerations about this manuscript. You have done a good job on this review.

Best Regards

Reviewer #3: Thank you for the opportunity to review your manuscript. You have undertaken a study to evaluate the content validity and feasibility of your eVIS intervention prior to undertaking an effectiveness trial.

This is a very comprehensive description of the work you’ve done to test and improve your intervention, in both a pre-clinical and a clinical setting. Your methods and results are quite clear, and I feel you have addressed the detail and flow issues raised by the previous reviews. I only have a few very minor points for your consideration.

Minor comments:

1. Page 5, Line 83. You seem to have a duplicated sentence here.

2. Page 5, line 90. “activity prescriptions based on individual hinders…”. I’m not sure what a ‘hinder’ is. Do you mean barrier? You may wish to change this word so it is consistent with the rest of your text.

3. Page 11, line 226. You have noted here that “further expert interviews were needed regarding two areas emerging as incomplete in previous assessment processes” Are you able to mention here what these two areas were here? Was it related to the pharmaceutical issues?

4. Table 1 – you have put xxx and x in the table. Can you add a description of what the different number of x’s mean in the table legend? I assume it’s the number of rounds they participated in?

7. PLOS authors have the option to publish the peer review history of their article (what does this mean?). If published, this will include your full peer review and any attached files.

Reviewer #1: No

Reviewer #3: **Yes: **Jocelyn Bowden

---

## [Author Response · Author response to Decision Letter 1]

1 Feb 2023

Reviewer Comments - point-by-point response.

Please note that indicated pages and lines refer to the revised paper without tracked changes, the file labeled 'Manuscript'.

Reviewer #1: 

Dear Authors,

I have no further considerations about this manuscript. You have done a good job on this review.

Best Regards

Author response

Thank you for your kind response. We are grateful for the time and effort you have put into evaluating our manuscript and your provided guidance for improvement. 

Reviewer #3: 

Thank you for the opportunity to review your manuscript. You have undertaken a study to evaluate the content validity and feasibility of your eVIS intervention prior to undertaking an effectiveness trial.

This is a very comprehensive description of the work you’ve done to test and improve your intervention, in both a pre-clinical and a clinical setting. Your methods and results are quite clear, and I feel you have addressed the detail and flow issues raised by the previous reviews. I only have a few very minor points for your consideration.

Minor comments:

1. Page 5, Line 83. You seem to have a duplicated sentence here.

Author response

Thank you for noticing. We have now deleted the duplicated sentence. Page 5, Line 83.

2. Page 5, line 90. “activity prescriptions based on individual hinders…”. I’m not sure what a ‘hinder’ is. Do you mean barrier? You may wish to change this word so it is consistent with the rest of your text.

Author response

Thank you for your suggestion, of course it should be “barrier”. We have now replaced the word hinders with barriers. Page 5, line 88. 

3. Page 11, line 226. You have noted here that “further expert interviews were needed regarding two areas emerging as incomplete in previous assessment processes” Are you able to mention here what these two areas were here? Was it related to the pharmaceutical issues?

Author response

Thank you, this is a good point. We have now clarified which the two areas were: recruitment logistics and pharmaceutical report function. Page 11, line 225.

4. Table 1 – you have put xxx and x in the table. Can you add a description of what the different number of x’s mean in the table legend? I assume it’s the number of rounds they participated in?

Author response

Thank you for your kind suggestions for improvement. We have now extended the table legend for a clearer explication of what is detailed in the table. We have also added a symbol explanation of the “x” in the footnote. Page 18-19, Table 1, line 2 and 415.

THANK YOU!

---

## [Decision Letter · Decision Letter 2]

23 Feb 2023

Evaluation of content validity and feasibility of the eVISualisation of physical activity and pain (eVIS) intervention for patients with chronic pain participating in interdisciplinary pain rehabilitation programs

PONE-D-22-23979R2

Dear Dr. Tseli,

We’re pleased to inform you that your manuscript has been judged scientifically suitable for publication and will be formally accepted for publication once it meets all outstanding technical requirements.

Kind regards,

Alejandro Vega-Muñoz, Ph.D.

Academic Editor

PLOS ONE

Additional Editor Comments (optional):

Reviewers' comments:

Reviewer's Responses to Questions

**Comments to the Author**

1. If the authors have adequately addressed your comments raised in a previous round of review and you feel that this manuscript is now acceptable for publication, you may indicate that here to bypass the “Comments to the Author” section, enter your conflict of interest statement in the “Confidential to Editor” section, and submit your "Accept" recommendation.

Reviewer #3: All comments have been addressed

2. Is the manuscript technically sound, and do the data support the conclusions?

Reviewer #3: Yes

3. Has the statistical analysis been performed appropriately and rigorously? 

Reviewer #3: Yes

4. Have the authors made all data underlying the findings in their manuscript fully available?

Reviewer #3: Yes

5. Is the manuscript presented in an intelligible fashion and written in standard English?

Reviewer #3: Yes

6. Review Comments to the Author

Reviewer #3: I have no further comments on this manuscript. All comments have been addressed.

7. PLOS authors have the option to publish the peer review history of their article (what does this mean?). If published, this will include your full peer review and any attached files.

Reviewer #3: **Yes: **Jocelyn Bowden

---

## [Editor Report · Acceptance letter]

28 Feb 2023

PONE-D-22-23979R2 

Evaluation of content validity and feasibility of the eVISualisation of physical activity and pain (eVIS) intervention for patients with chronic pain participating in interdisciplinary pain rehabilitation programs 

Dear Dr. Tseli:

I'm pleased to inform you that your manuscript has been deemed suitable for publication in PLOS ONE. Congratulations! Your manuscript is now with our production department. 

Kind regards, 

on behalf of

Dr. Alejandro Vega-Muñoz 

Academic Editor

PLOS ONE